# Does Delaying Time in Cancer Treatment Affect Mortality? A Retrospective Cohort Study of Korean Lung and Gastric Cancer Patients

**DOI:** 10.3390/ijerph18073462

**Published:** 2021-03-26

**Authors:** Kyu-Tae Han, Woorim Kim, Seungju Kim

**Affiliations:** 1Division of Cancer Control and Policy, National Cancer Control Institute, National Cancer Center, Goyang 10408, Korea; kthan.phd@gmail.com (K.-T.H.); wklaura@gmail.com (W.K.); 2Department of Nursing, College of Nursing, The Catholic University of Korea, Seoul 06591, Korea

**Keywords:** time-to-treatment, lung cancer, cancer care, mortality, regional disparity

## Abstract

The aim of this study is to investigate the association between delays in surgical treatment and five- and one- year mortality in patients with lung or gastric cancer. The National Health Insurance claims data from 2006 to 2015 were used. The association between time to surgical treatment, in which the cut-off value was set at average time (30 or 50 days), and five year mortality was analyzed using the Cox proportional hazard model. Subgroup analysis was performed based on treatment type and location of medical institution. A total of 810 lung and 2659 gastric cancer patients were included, in which 74.8% of lung and 71.2% of gastric cancer patients received surgery within average. Compared to lung cancer patients who received treatment within 50 days, the five-year (HR 1.826, 95% CI 1.437–2.321) mortality of those who received treatment afterwards was higher. The findings were not significant for gastric cancer based on the after 30 days standard (HR: 1.003, 95% CI: 0.822–1.225). In lung cancer patients, time-to-treatment and mortality risk were significantly different depending on region. Delays in surgical treatment were associated with mortality in lung cancer patients. The findings imply the importance of monitoring and assuring timely treatment in lung cancer patients.

## 1. Introduction

Lung and gastric cancer are a significant health problem in numerous countries, including South Korea. Lung cancer is a leading cause of death worldwide and Korea is no exception, with its age standardized mortality rate for lung cancer being 16.7 per 100,000 people in year 2017 [1,2]. Compared to the 2013 to 2017 average cancer survival rate of around 70% in Korea, the survival rate of lung cancer is rated far below at only 30.2% [2]. As for gastric cancer, it is the fifth most common cancer and the third leading cause of cancer deaths globally [3]. Although the occurrence of gastric cancer is decreasing in many developed countries, issues around gastric cancer are important in Korea as it ranks first in incidence and fourth in mortality related to cancer [2]. Therefore, early detection, as well as timing of treatment, are important in reducing the burden of cancer patients and improving patient outcomes.

An important factor in investigating overall cancer survival is time-to-treatment, widely considered as an important quality indicator for cancer care [4]. In fact, postponements in time-to-treatment in cancer are known to commonly incur distress in patients and have been linked to oncologic outcomes [5,6]. Previous findings have shown that treatment delays are related to survival in certain types of cancer, such as lung cancer [7]. Likewise, a study conducted in the United States of America (USA) concluded an association between time-to-treatment delay and increased absolute mortality risk in curative settings, whilst other studies suggest no significant findings [6,8]. Regarding lung cancer, timeliness of care is recommended in many clinical practice guidelines, including that of the British National Health Service (NHS) and the USA [9]. Studies on the effect of time-to-treatment initiation and mortality in Korean lung cancer patients show mixed results, with both statistically significant and non-significant correlations being reported [10,11]. Studies on this subject in gastric cancer patients are less common, although a previous study completed in the Netherlands concluded no significant association between longer waiting time and overall survival [4].

The healthcare delivery system of Korea encompasses primary, secondary, and tertiary hospitals, with relatively large hospitals being largely concentrated in the capital area. Differences in access to care may be one of the many factors that affect patient care, and patients living away from hospitals report having worse outcomes, including mortality [12,13]. This means that region, such as urban or rural areas, can affect the timing of treatment. In addition, patient outcomes may differ depending on the type of treatment they receive. Therefore, a need exists to evaluate the impact of treatment delay on patient outcomes based on the location of the medical institution and the type of treatment received.

In this study, the association between time to surgical treatment and mortality in patients with gastric or lung cancer was investigated. We hypothesized that delayed treatment can reduce survival, especially in lung cancer patients with poor prognosis. Furthermore, a subgroup analysis was performed to evaluate the impact of treatment type and the location of the medical institution first visited for treatment on survival rate as these factors can affect patient outcomes.

## 2. Methods

### 2.1. Study Population

The 2002 to 2015 National Health Insurance (NHI) Cohort Database was used in this study. These data included a total of 1,000,000 people, sampled based on all registered residents in Korea at 2006 (*n* = 48,222,537). The data included information on socioeconomic status, treatment, medical examinations, and medical institutions.

As this study aimed to investigate the association between time to surgical treatment and outcomes in lung or gastric cancer patients, a total of 27,579 patients diagnosed with gastric or lung cancer were included at baseline. A wash-out period of three years was applied to clearly define the date of initial cancer diagnosis. In addition, patients who were diagnosed with other cancer types during the follow-up period were excluded. In this process, 8340 patients were excluded. To reduce heterogeneity among patients, only cancer patients who received surgical treatment within one year of diagnosis were included. Individuals who had died within 30 days of first diagnosis were excluded to avoid bias regarding the time to event outcome. Elderly patients over 80 years of age or Medical-Aid beneficiaries were also excluded. This led to the final study population of 810 and 2659 patients who received surgical treatment for lung or gastric cancer (Appendix A).

### 2.2. Variables

The outcome variable in this study was survival. Survival time was defined as the period between the end time (death or censorship) from the first diagnosis of cancer. Date of first diagnosis was defined as the first date of each patient’s visit to the hospital for lung or gastric cancer (International Statistical Classification of Diseases and Related Health Problems [ICD]-10: C33–34 or C16). Individuals were observed for up to five years, and those who died within one to five years of initial diagnosis were categorized into the “death” group.

The interesting variable was time to surgical treatment, referring to the period between the date of surgical treatment to the date of initial diagnosis. Categorization was based on the average time for each type of cancer (50 days in lung cancer, 30 days in gastric cancer). 

Other independent variables in this study were type of treatment; type or location of medical institution at first treatment; sex; age (less than 49 years, 50 to 54, 55 to 59, 60 to 64, 65 to 69, 70 to 74, or more than 75 years; income level (20, 40, 60, 80, 81+ percentiles); type of insurance coverage; residential area (capital area, metropolitan, or others); and the Charlson Comorbidity Index (CCI). First, individuals were divided into only surgical or with other treatments (chemotherapy and radiotherapy) groups. Medical institution at first treatment was categorized based on type (general hospital or others) and region (capital area, metropolitan, or others). Regarding insurance coverage, about 97% of individuals were National Health Insurance (NHI) beneficiaries and were classified into NHI self-employed and employee groups, and the remaining 3% were Medical Aid beneficiaries. The NHI employee group consisted of all employees and employers and their household members. The NHI self-employed group included all other individuals. Premiums were calculated according to income, property, and living standards. The Medical Aid group included about 3% of low-income or disabled individuals and did not require premiums, and this population was excluded in this study. During the first year of diagnosis, CCI was used to integrate clinical severity calculated based on medical records, excluding cancer (without cancer; 3 or less, 4 to 6, or 7 or above).

### 2.3. Statistical Analysis

The general characteristics of the study population were analyzed using chi-square tests for categorical variables and *t*-tests for continuous variables. We presented the results on average days in Figure 1 to assess the period of time delay from diagnosis by region and study period. Kaplan–Meier survival curves and log-rank test were used to compare survival rates based on time to surgical treatment from diagnosis based on average time (30 or 50 days). Next, survival analysis using the Cox proportional hazard model were conducted after controlling for all independent variables to investigate the association between time to treatment and survival at 5 years of diagnosis. Subgroup analyses according to type of treatment and location of first treatment were conducted to compare the differences between groups. All statistical analyses were performed using the SAS statistical software version 9.4 (Cary, NC, USA). A *p*-value < 0.05 was considered statistically significant.

## 3. Results

Table 1 presents the baseline characteristics of the study population. The percentage of individuals who received treatment within 50 days was higher for lung cancer as 606 (74.8%) participants received treatment within 50 days and 204 (25.2%) after 50 days. Among gastric cancer patients, 1893 (71.2%) individuals received treatment within 30 days and 766 individuals after 30 days. In the case of lung cancer, 78.4% of patients living in the capital area, 68.4% in the metropolitan area, and 75.2% in other regions received surgery within 50 days (*p* = 0.0411). As for treatment type, 71.5% of patients who only received surgery and 79.2% of patients who combined surgery with other treatment regimens received surgery within 50 days (*p* = 0.0119). Depending on the location of the medical institution visited for initial treatment, 80.4% of patients treated in hospitals located in the capital area received surgery within 50 days. This figure was 69.4% for patients in the metropolitan area and 68.7% for those in other areas (*p* = 0.0010). In the case of gastric cancer, 69.5% of male and 74.7% of female patients received surgery within 30 days, in which the results were statistically significant. Regarding type of treatment, 75.1% of patients who only received surgery and 69.4% of patients who received surgery and other treatment regimens underwent surgery within 30 days (*p* = 0.0029).

Figure 1 presents the national trend of time-to-treatment in the study participants. The average period of time-to-treatment in lung cancer patients was shortest in patients living in the capital area (45.7 days) and longest in patients at other areas (54.2 days). Over the past 10 years, the average time-to-treatment for lung cancer patients decreasing in all regions. In the case of gastric cancer, unlike lung cancer, the average period of time-to-treatment was longest in the capital area. Overall, treatment time had declined in the past 10 years.

Figure 2 shows the results of the survival curves and the log-rank test. Lung cancer patients who received surgical treatment within 50 days had higher survival rates than those treated after 50 days (1449.9 days versus 1164.7, *p*-value < 0.0001). In gastric cancer patients, no statistically significant differences were found (1638.9 days versus 1645.2 days, *p*-value = 0.3721).

Table 2 shows the results of the analysis on the association between time to surgical treatment and mortality. Compared to lung cancer patients who received treatment within 50 days, the 5-year mortality risk of those who received surgery after 50 days (HR 1.826, 95% CI 1.437–2.321) was higher. Although similar tendency in 5-year mortality was seen in gastric cancer patients, the findings did not show statistical significance. Risk of mortality increased with age in lung cancer patients, but only with statistical significance in patients aged over 75 years (HR: 2.932, 95% CI: 1.513–5.681). No significant association was found between socioeconomic factors such as income and insurance and risk of death in lung cancer patients. There was a significant increase in mortality in patients with a CCI score of 7 or higher (HR: 2.114, 95% CI: 1.436–3.112) or who underwent surgery plus other treatment (HR: 3.884, 95% CI: 2.836–5.320). Similar results were shown in patient with gastric cancer and patients with a high CCI score showed increased risk of mortality (CCI 4–6 HR: 1.428, 95% CI: 1.071–1.903; 7+ HR: 2.950, 95% CI: 2.267–3.838). Patients who underwent surgery plus other treatment had a higher risk of mortality than patients who underwent surgery only (HR: 5.800, 95% CI: 4.710–7.142).

Figure 3 reveals the results of the subgroup analysis. The trends shown in the main analysis was generally maintained. Interestingly, the higher risk of 5-year mortality seen among lung cancer patients experiencing treatment delays were statistically maintained in patients who received only surgical treatment. Higher 5-year mortality risks found in lung cancer patients who received treatment after 50 days were maintained in the non-capital-area group (metropolitan HR: 1.792, 95% CI: 1.070–3.002; others HR: 2.212, 95% CI: 1.472–3.322).

## 4. Discussion

The results of this study reveal that time to surgical treatment is associated with mortality in lung cancer patients, as those who received treatment after 50 days had a higher risk of 5-year mortality. Similar tendencies were found in gastric cancer patients but without statistical significance. Interestingly, this association found in lung cancer patients was particularly noticeable in those who underwent only surgery and received treatment in the non-capital area.

Whilst previous studies report both significant and non-significant correlations between time to treatment and mortality in lung cancer patients, the findings of this study add evidence that delays in surgical treatment may increase mortality risk [14,15]. Such tendencies may result as delays in initiating treatment and may lead to tumor progression and poorer prognosis, affecting mortality risk [16]. Studies stating otherwise report that patients with advanced disease generally experience shorter delays, which in turn can result in no association between hospital delays and survival [17]. However, these studies often constituted a population in which a large proportion experienced treatment delays, with only around 30% of individuals reporting to undergo surgery within six weeks or had a long median delay from diagnosis to treatment of 53 days [17,18]. Furthermore, studies also tended to include many patients at a later stage unfit for resections, which will likely result in shortened delays [18]. Since this study only included first diagnosed patients who underwent surgery in 1 year and did not die within 30 days of diagnosis, the findings are noteworthy as it infer that surgical treatment delay correlated with increased 5-year mortality risk in lung cancer patients. 

The correlation between treatment delay and mortality was statistically non-significant in gastric cancer patients. Compared to lung cancer, this subject was relatively less studied in gastric cancer. One cohort study conducted in the Netherlands concluded that longer waiting times are not related to overall survival [4]. A study on Korean patients also revealed similar results, with no statistically significant association being found between surgical treatment delay and 5-year survival [10]. Another Korean study reported only an optimal time interval of 4 weeks for adjuvant chemotherapy in gastric cancer patients who received surgery [19]. The tendencies may be a reflection of the comparatively high survival rate of gastric cancer, which reached 76.5% between 2013 to 2017 whereas that of lung cancer was 30.2% for the identical time period [2]. Furthermore, the proportion of patients diagnosed at earlier stages have increased, with this figure reaching over 50%, leading to more frequent application of minimally invasive patients [20,21]. In fact, the proportion of gastric cancer patients experiencing surgical treatment delays of over 30 days was also noticeably lower than lung cancer patients in this study.

The results of the subgroup analysis reveal that type of treatment and the of the visiting medical institution can influence patient outcome regarding lung cancer. Lung cancer survival is largely determined by stage and treatment characteristics. Surgical resection is commonly conducted in lower stage patients without metastasis whilst later stage operable patients often receive pre- or post-chemo and/or radiotherapy [22]. Previous studies have report that delayed resections may reduce survival rates [7] and hence suggest the importance of timely surgery [23]. Our results also suggest that it is important to receive surgery on time, especially if only surgical treatment is needed, as timely treatment may affect patient outcomes.

In addition, the effect of time to surgical treatment on mortality was more pronounced in patients who received treatment in the non-capital area. This phenomenon can be considered in two aspects. First, the regional imbalance of medical resources must be taken into account [24]. In Korea, most tertiary hospitals are concentrated in the capital area, which are equipped with various professional staff and high-quality radiotherapy facilities and provide high-quality healthcare services [25]. Such tendencies can act affect the patient’s choice, which is the second aspect. Since no strong referral system is in place, patients prefer relatively large hospitals based on individual beliefs [26]. In fact, studies show that the number of surgeries performed for lung cancer has increased in large cities where tertiary hospitals are located [25]. In particular, 60% of lung cancer surgeries were conducted in tertiary hospitals located in the capital area and most patients were receiving radiation therapy in Seoul [25,27]. In terms of healthcare resources, well-equipped medical equipment and physician volume can lead to better patient outcomes [28], which in turn can affect mortality rates.

These results suggest that while differences in treatment time between regions have decreased over the past decade, health disparities still exist as a potential problem. In particular, differences in access to care across regions can lead to delays in treatment, which can increase risk of mortality. Therefore, policy makers should take into consideration patient accessibility, in particular for patients living in vulnerable areas so that medical resources can be more evenly distributed across regions. In addition, health gaps between regions must be continuously considered through routine evaluation of health outcomes between regions.

This study is not without its limitations. First, cancer stage could not be adjusted in the analysis due to data limitation. To overcome this limitation, the study population was selected to include only first diagnosed lung or gastric cancer patients who underwent surgery in one year and did not die within 30 or 50 days of diagnosis. The type of treatment patients received was also included as a covariate. Second, although the analysis accounted for various demographic, social, and health related variables, the potential possibility of residual confounding cannot be entirely ruled out. Information on smoking and alcohol behaviors could not be considered due to data limitation. Third, variables on lung function testing were also not available. Finally, the study results may be affected by the cutoff criterion and further studies that consider various waiting times are needed. Despite such limitations, this study is noteworthy as it investigated the relationship between delays in surgical treatment and mortality in lung and gastric cancer patients using a large, nationally representative sample of individuals.

## 5. Conclusions

Delays in surgical treatment were associated with 5-year mortality in lung cancer patients. Similar tendencies were found in gastric cancer patients, but without statistical significance. The correlations were comparatively pronounced in patients who had received only surgical treatment and visited institutions located in non-capital areas. The findings imply the importance of timely treatment in lung cancer patients.

## Figures and Tables

**Figure 1 ijerph-18-03462-f001:**
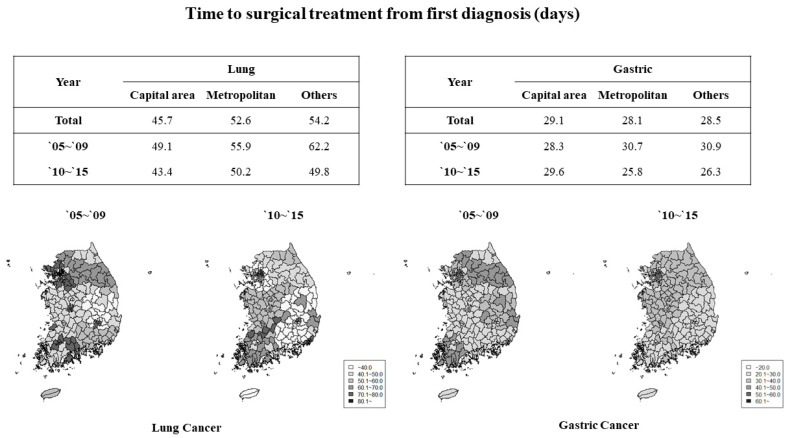
Average time-to-treatment of the study participants by region (days).

**Figure 2 ijerph-18-03462-f002:**
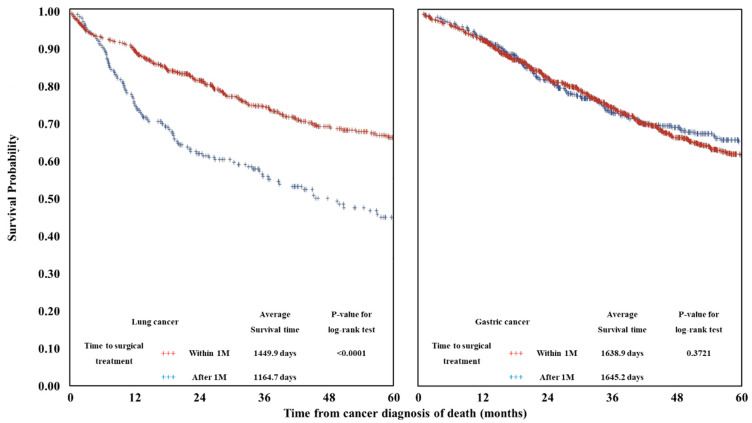
Survival curve and results of log rank test.

**Figure 3 ijerph-18-03462-f003:**
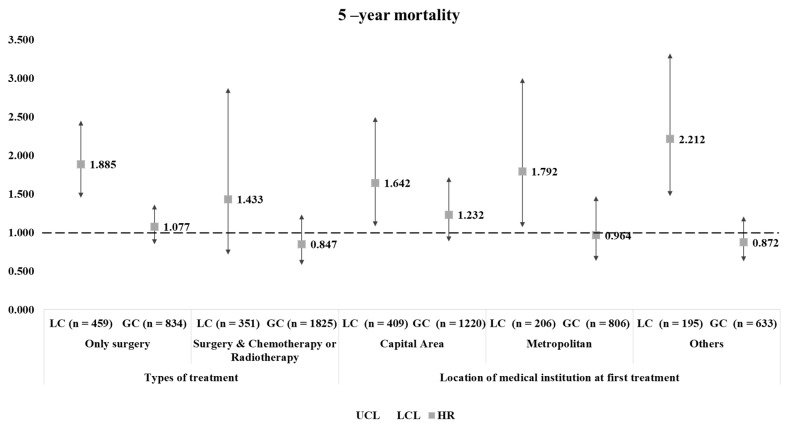
Results of the subgroup analysis on the association between time to surgical treatment and mortality.

**Table 1 ijerph-18-03462-t001:** Baseline characteristics of cancer patients with surgical treatment.

Variables	Lung Cancer					Gastric Cancer				
	Within 50 Days		After 51 Days		*p*-Value	Within 30 Days		After 31 Days		*p*-Value
	N/Mean	%/SD	N/Mean	%/SD		N/Mean	%/SD	N/Mean	%/SD	
Sex										
Male	409	73.7	146	26.3	0.278	1252	69.5	549	30.5	0.006
Female	197	77.3	58	22.7		641	74.7	217	25.3	
Age (Years)										
~49	40	80.0	10	20.0	0.500	221	72.5	84	27.5	0.145
50~54	29	67.4	14	32.6		201	74.4	69	25.6	
54~59	50	70.4	21	29.6		251	71.3	101	28.7	
60~64	101	73.7	36	26.3		278	75.3	91	24.7	
65~69	109	80.7	26	19.3		257	67.6	123	32.4	
70~74	112	76.7	34	23.3		274	67.7	131	32.3	
75+	165	72.4	63	27.6		411	71.1	167	28.9	
Income Level										
~20 Percentile	72	66.7	36	33.3	0.066	286	72.8	107	27.2	0.196
21~40 Percentile	68	70.8	28	29.2		274	74.1	96	25.9	
41~60 Percentile	105	77.8	30	22.2		338	69.5	148	30.5	
61~80 Percentile	138	72.6	52	27.4		424	74.0	149	26.0	
81 Percentile~	223	79.4	58	20.6		571	68.2	266	31.8	
Types of Insurance Coverage										
NHI, Self-employed	192	71.6	76	28.4	0.144	671	71.0	274	29.0	0.875
NHI, Employee	414	76.4	128	23.6		1222	71.3	492	28.7	
Residence Area										
Capital area	250	78.4	69	21.6	0.041	686	70.9	282	29.1	0.740
Metropolitan	132	68.4	61	31.6		512	72.3	196	27.7	
Others	224	75.2	74	24.8		695	70.7	288	29.3	
Charlson Comorbidity Index										
~3	136	81.0	32	19.0	0.058	703	74.9	236	25.1	0.005
4~6	202	75.7	65	24.3		660	68.1	309	31.9	
7~	268	71.5	107	28.5		530	70.6	221	29.4	
Types of Treatment										
Only surgery	328	71.5	131	28.5	0.012	626	75.1	208	24.9	0.003
Surgery & Chemotherapy or Radiotherapy	278	79.2	73	20.8		1267	69.4	558	30.6	
Types of Medical Institution at First Treatment										
General Hospital	217	81.3	50	18.7	0.003	546	69.1	244	30.9	0.124
Others	389	71.6	154	28.4		1347	72.1	522	27.9	
Location of Medical Institution at First Treatment										
Capital area	329	80.4	80	19.6	0.001	859	70.4	361	29.6	0.711
Metropolitan	143	69.4	63	30.6		580	72.0	226	28.0	
Others	134	68.7	61	31.3		454	71.7	179	28.3	
Total	606	74.8	204	25.2		1893	71.2	766	28.8	

N: number; SD: Standard Deviation; The *p*-value is the result of the Chi-square test or the result of the ANOVA test.

**Table 2 ijerph-18-03462-t002:** Results of survival analysis by time to surgical treatment.

Variables	Lung Cancer				Gastric Cancer			
	HR	95% CI		*p*-Value	HR	95% CI		*p*-Value
Time to Surgical Treatment								
Below average time	1.000	-	-	-	1.000	-	-	-
Above average time	1.826	1.437	2.321	<0.001	1.003	0.822	1.225	0.974
Sex								
Male	1.280	0.982	1.669	0.068	1.142	0.935	1.396	0.193
Female	1.000	-	-	-	1.000	-	-	-
Age (Years)								
~49	1.000	-	-	-	1.000	-	-	-
50~54	0.975	0.416	2.287	0.954	0.621	0.390	0.991	0.046
54~59	1.197	0.559	2.563	0.643	0.671	0.441	1.021	0.062
60~64	1.416	0.709	2.831	0.325	0.788	0.515	1.206	0.272
65~69	1.356	0.681	2.700	0.386	0.849	0.572	1.261	0.418
70~74	1.271	0.639	2.530	0.495	1.230	0.852	1.774	0.269
75+	2.932	1.513	5.681	0.001	2.216	1.571	3.125	<0.001
Income Level								
~20 Percentile	0.813	0.526	1.257	0.351	0.818	0.591	1.132	0.226
21~40 Percentile	0.793	0.525	1.198	0.270	0.822	0.611	1.107	0.197
41~60 Percentile	0.750	0.512	1.098	0.139	0.721	0.540	0.962	0.026
61~80 Percentile	0.773	0.536	1.113	0.166	0.768	0.584	1.010	0.059
81 Percentile~	1.000	-	-	-	1.000	-	-	-
Types of Insurance Coverage								
NHI, Self-employed	1.212	0.956	1.536	0.112	1.082	0.897	1.306	0.410
NHI, Employee	1.000	-	-	-	1.000	-	-	-
Residence Area								
Capital area	1.336	0.848	2.104	0.212	1.234	0.844	1.803	0.278
Metropolitan	1.371	0.917	2.049	0.124	0.879	0.634	1.218	0.439
Others	1.000	-	-	-	1.000	-	-	-
Charlson Comorbidity Index								
~3	1.000	-	-	-	1.000	-	-	-
4~6	1.201	0.789	1.829	0.393	1.428	1.071	1.903	0.015
7~	2.114	1.436	3.112	0.0001	2.950	2.267	3.838	<0.001
Types of Treatment								
Only surgery	1.000	-	-	-	1.000	-	-	-
Surgery & Chemotherapy or Radiotherapy	3.884	2.836	5.320	<0.001	5.800	4.710	7.142	<0.001
Types of Medical Institution at First Treatment								
General Hospital	1.059	0.745	1.505	0.750	0.813	0.611	1.082	0.156
Others	1.000	-	-	-	1.000	-	-	-
Location of Medical Institution at First Treatment								
Capital area	1.000	-	-	-	1.000	-	-	-
Metropolitan	0.962	0.600	1.540	0.870	0.769	0.513	1.152	0.203
Others	0.934	0.582	1.497	0.776	1.117	0.757	1.648	0.577

This is the result of survival analysis, and all variables were included simultaneously.

## Data Availability

Data is available from NHIS upon request.

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
