# Peer review of "Does Delaying Time in Cancer Treatment Affect Mortality? A Retrospective Cohort Study of Korean Lung and Gastric Cancer Patients"

_ijerph, 2021, doi:10.3390/ijerph18073462_

Round 1

Reviewer 1 Report

In the paper submitted by Kim et al., authors investigate the relation between a delayed surgical treatments and mortality of patients affected by lung or gastric cancer using statistical analyses of data extracted from the National Health Insurance Cohort Databases in the period 2002-2015 in Korea.

The results of this analysis, obtained using standard statistical techniques, show that a positive and significant correlation between a delayed surgical treatment and mortality exists in lung cancer patients. For gastric cancer patients the trend is similar to the previous one but not statistical significance was observed.

The idea of the paper is good but I have some concerns about the selection of the database, the analyses performed and the presentation of the paper.

Database selection

The authors started from a very large and complete database but they selected a relatively small sample in order to avoid confounding effects on their results. All the information contained in the whole database was lost.

I wonder if the authors have analysed the entire database and have proved that their choice was the right one. I can suppose that they have eliminate confounding effects but have reduced a lot the heterogeneity of the population under study reducing the power of the analyses.

A proper description of the statistical characteristics of the entire database should be discussed and, if the authors have them, a comparison of the results obtained from the entire sample with their small sample could give many interesting indications and could demonstrate the accuracy of their selection.

Analyses performed and presentation of the paper

I think that there is a huge pagination problem as lots line break splitted words are present throughout the text. I report here only the ones found in introduction as example: av-erage line 34, Alt-hough line 37, fea-ture line 39, find-ings in line 46, sig-nificance in line 50 and 56,sta-tistically in line 53 and so on. This obviously has to be corrected throughout all the text so the authors have to carefully read the text and correct this inconvenience.

I have found the results section too short and concise. The author have not presented the significant results in table 1 that it is also incorrect in its headers. I don’t understand the meaning of the headers: N/Mean and %/ SD mean as they are simply the sample size and its percentage on the total of the sample. This has to be corrected and a reference to the significance of the tests must be given in the text and their impact on the following analyses has to be also explained.

Table 2 presents the same concerns expressed previously for table 1. The authors have to explain or present the significant results and their impact on the following analyses.

Figure 2 and subgroup analysis. Figure 2 is a very compact figure and the subgroup sample sizes is not presented. In subgroup analysis the sample size change a lot as many subgroups are subdivided and this influences the strength of the results. Moreover no statistical results is given and no discussion is presented. This have to be changed.

Figure 3. Again this figure show results with no statistical test. It is only a visual representation of the data: indicative but only descriptive.

The paper is very simple and, in my opinion, if the discussion were reduced, it will perfectly fit in a “short communication” section of your Journal. As a proper article, it must be improved. I think it has to be expanded, adding other analyses such as multivariate statistical analysis which takes into account all the variables at the same time. The paper should be also improved, balancing the different paragraphs of the paper. I am sure that the authors did much more analyses than the ones they have presented and I think they have to review them and make a new selection to increase the reliability of their results. In my opinion, statistics provide many techniques that can help pointing out interesting results and a balance between too many analyses, sometimes misleading, and too few, not sufficient to corroborate an analysis, has to be reached.

In view of these considerations, I think that the paper can be reconsidered after revisions.

Author Response

First, we greatly appreciate the comments and suggestions offered by the reviewers, which we used to improve the manuscript. Our response to each comment follows, and we have attached a revision note and also highlighted the revised sections of the manuscript. Again, thank you for the valuable and helpful comments.

Answer to Reviewer 1:

  1. The authors started from a very large and complete database but they selected a relatively small sample in order to avoid confounding effects on their results. All the information contained in the whole database was lost. I wonder if the authors have analysed the entire database and have proved that their choice was the right one. I can suppose that they have eliminate confounding effects but have reduced a lot the heterogeneity of the population under study reducing the power of the analyses. A proper description of the statistical characteristics of the entire database should be discussed and, if the authors have them, a comparison of the results obtained from the entire sample with their small sample could give many interesting indications and could demonstrate the accuracy of their selection.

 Answer: Thank you for your comments.

The data we used included approximately one million participants, of which 27,579 patients diagnosed with gastric or lung cancer were included in the baseline participants. We excluded the 3-year period (wash-out period) for a clear definition of the initial diagnosis, of which only patients who underwent surgery were included. In addition, patients with factors that may affect the patient's outcome, such as death within 30 days of cancer diagnosis, age and reoperation within 1 years, were excluded. We added a flow diagram of the study population and revised the method section (page2 line 76-86).

A total of 27,579 patients diagnosed with gastric or lung cancer were included at baseline. A wash out period of three years was applied to clearly define the date of initial cancer diagnosis. In addition, patients who were diagnosed with other cancer types during the follow-up period were excluded. In this process, 8,340 patients were excluded. To reduce heterogeneity among patients, only cancer patients who received surgical treatment within one year of diagnosis were included. Individuals who died within 30 days of first diagnosis were excluded to avoid bias regarding the time to event outcome. Elderly patients over 80 years of age or Medical-Aid beneficiaries were also excluded. This led to the final study population of 810 and 2,659 patients who received surgical treatment for lung or gastric cancer (Appendix figure 1).

  1. Analyses performed and presentation of the paper

I think that there is a huge pagination problem as lots line break splitted words are present throughout the text. I report here only the ones found in introduction as example: av-erage line 34, Alt-hough line 37, fea-ture line 39, find-ings in line 46, sig-nificance in line 50 and 56,sta-tistically in line 53 and so on. This obviously has to be corrected throughout all the text so the authors have to carefully read the text and correct this inconvenience.

 Answer: Thank you for your comment. The split words appear to have occurred during the journal formatting process. We have corrected these words throughout the manuscript.

  1. I have found the results section too short and concise. The author have not presented the significant results in table 1 that it is also incorrect in its headers. I don’t understand the meaning of the headers: N/Mean and %/ SD mean as they are simply the sample size and its percentage on the total of the sample. This has to be corrected and a reference to the significance of the tests must be given in the text and their impact on the following analyses has to be also explained.

 Answer: Thank you for your comments. We have added the Table 1 results to the results section as follows (page 3 line 134-145):

Table 1 presents the baseline characteristics of the study population. The percentage of individuals who received treatment within 50 days was higher for lung cancer as 606 (74.8%) participants received treatment within 50 days and 204 (25.2%) after 50 days. Among gastric cancer patients, 1,893 (71.2%) individuals received treatment within 30 days and 766 individuals after 30 days. In the case of lung cancer, 78.4% of patients living in the capital area, 68.4% in the metropolitan area, and 75.2% in other regions received surgery within 50 days (p=0.0411). As for treatment type, 71.5% of patients who only received surgery and 79.2% of patients who combined surgery with other treatment regimens received surgery within 50 days (p=0.0119). Depending on the location of the medical institution visited for initial treatment, 80.4% of patients treated in hospitals located in the capital area received surgery within 50 days. This figure was 69.4% for patients in the metropolitan area and 68.7% for those in other areas (p=0.0010). In the case of gastric cancer, 69.5% of male and 74.7% of female patients received surgery within 30 days, in which the results were statistically significant. Regarding type of treatment, 75.1% of patients who only received surgery and 69.4% of patients who received surgery and other treatment regimens underwent surgery within 30 days (p=0.0029).

And added the full term of the abbreviation in Table 1 as a footnote (page 4).

N: number; SD: Standard Deviation

  1. Table 2 presents the same concerns expressed previously for table 1. The authors have to explain or present the significant results and their impact on the following analyses.

 Answer: Thank you for your comment. We have added the Table 2 results to the results section as follows (page 3 line 168-177):

Table 2 shows the results of the analysis on the association between time to surgical treatment and mortality. Compared to lung cancer patients who received treatment within 50 days, the five-year mortality risk of those who received surgery after 50 days (HR 1.826, 95% CI 1.437 - 2.321) was higher. Although similar tendency in five- year mortality was seen in gastric cancer patients, the findings did not show statistical significance. Risk of mortality increased with age in lung cancer patients, but only with statistical significance in patients aged over 75 years (HR: 2.932, 95% CI: 1.513-5.681). No significant association was found between socioeconomic factors such as income and insurance and risk of death in lung cancer patients. There was a significant increase in mortality in patients with a CCI score of 7 or higher (HR: 2.114, 95% CI: 1.436-3.112) or who underwent surgery plus other treatment (HR: 3.884, 95% CI: 2.836-5.320). Similar results were shown in patient with gastric cancer and patients with a high CCI score showed increased risk of mortality (CCI 4-6 HR: 1.428, 95% CI: 1.071-1.903; 7+ HR: 2.950, 95% CI: 2.267-3.838). Patients who underwent surgery plus other treatment had a higher risk of mortality than patients who underwent surgery only (HR: 5.800, 95% CI: 4.710-7.142).

  1. Figure 2 and subgroup analysis. Figure 2 is a very compact figure and the subgroup sample sizes is not presented. In subgroup analysis the sample size change a lot as many subgroups are subdivided and this influences the strength of the results. Moreover no statistical results is given and no discussion is presented. This have to be changed.

 Answer: Thank you for your comments.

We carefully considered the presentation of our results based on your comments (#5, #7), and decided to present our analysis on the subgroup only by type of treatment and location of medical institution. Type of treatment is a factor that is closely related to patient outcome and our findings provide evidence that timely treatment is important, especially for patients who only undergo surgical treatment. In addition, Korea has an imbalance in the distribution healthcare resources and because most large hospitals are concentrated in the capital area, different health outcomes are found depending on region. Our study shows that patients with gastric cancer have an increased risk of mortality if they do not receive surgery at the right time, and that the magnitude of these effects varies from region to region. These findings suggest that adequate time is important for surgery in cancer patients and hence, continued attention is needed to reduce the health gap between regions.

The sample sizes of the subgroups are presented in figure 2(= figure3). The detailed number of patients is given in Table 1.

Figure 3. Results of the subgroup analysis on the association between time to surgical treatment and mortality.

We revised introduction section as following (page 2 line 54-68):

The healthcare delivery system of Korea encompasses primary, secondary, and tertiary hospitals, with relatively large hospitals being largely concentrated in the capital area. Differences in access to care may be one of the many factors that affect patient care, and patients living away from hospitals report having worse outcomes, including mortality [12, 13]. This means that region, such as urban or rural areas, can affect the timing of treatment. In addition, patient outcome may differ depending on the type of treatment they receive. Therefore, a need exists to evaluate the impact of treatment delay on patient outcomes based on the location of the medical institution and the type of treatment received.

In this study, the association between time to surgical treatment and mortality in patients with gastric or lung cancer was investigated. We hypothesized that delayed treatment can reduce survival, especially in lung cancer patients with poor prognosis. Furthermore, a subgroup analysis was performed to evaluate the impact of treatment type and the location of the medical institution first visited for treatment on survival rate as these factors can affect patient outcomes.

  1. Figure 3. Again this figure show results with no statistical test. It is only a visual representation of the data: indicative but only descriptive.

 Answer: Thank you for your comment. As in the reviewer's comment, Figure 3 shows the average time-to-treatment, not results of a statistical analysis. Although there is no statistical analysis, we thought that it was meaningful to show the different treatment times by region using a map. We presented maps to help readers better understand the differences in time-to-treatment between regions. In addition, we changed Figure 3 to 1, which does not include the analysis taking into account statistical flow.

  1. The paper is very simple and, in my opinion, if the discussion were reduced, it will perfectly fit in a “short communication” section of your Journal. As a proper article, it must be improved. I think it has to be expanded, adding other analyses such as multivariate statistical analysis which takes into account all the variables at the same time. The paper should be also improved, balancing the different paragraphs of the paper. I am sure that the authors did much more analyses than the ones they have presented and I think they have to review them and make a new selection to increase the reliability of their results. In my opinion, statistics provide many techniques that can help pointing out interesting results and a balance between too many analyses, sometimes misleading, and too few, not sufficient to corroborate an analysis, has to be reached.

 Answer: Thank you for your comments.

The main result of our study is the results of the multivariate analysis in which all of the variables presented in Table 1 were included simultaneously. We have carefully considered presenting our results based on the reviewers' opinions and have revised the discussion section as follows (page 9 line 225-255):

The results of the subgroup analysis reveal that type of treatment and the of the visiting medical institution can influence patient outcome regarding lung cancer. Lung cancer survival is largely determined by stage and treatment characteristics. Surgical resection is commonly conducted in lower stage patients without metastasis whilst later stage operable patients often receive pre- or post- chemo and or radiotherapy [22]. Previous studies have report that delayed resections may reduce survival rates [7] and hence suggest the importance of timely surgery [23]. Our results also suggest that it is important to receive surgery on time, especially if only surgical treatment is needed, as timely treatment may affect patient outcomes.

In addition, the effect of time to surgical treatment on mortality was more pronounced in patients who received treatment in the non-capital area. This phenomenon can be considered in two aspects. First, the regional imbalance of medical resources must be taken into account [24]. In Korea, most tertiary hospitals are concentrated in the capital area, which are equipped with various professional staff and high-quality radiotherapy facilities and provide high-quality healthcare services [25]. Such tendencies can act affect the patient's choice, which is the second aspect. Since no strong referral system is in place, patients prefer relatively large hospitals based on individual beliefs [26]. In fact, studies show that the number of surgeries performed for lung cancer has increased in large cities where tertiary hospitals are located [25]. In particular, 60% of lung cancer surgeries were conducted in tertiary hospitals located in the capital area and most patients were receiving radiation therapy in Seoul [25, 27]. In terms of healthcare resources, well-equipped medical equipment and physician volume can lead to better patient outcomes [28], which in turn can affect mortality rates.

These results suggest that while differences in treatment time between regions have decreased over the past decade, health disparities still exist as a potential problem. In particular, differences in access to care across regions can lead to delays in treatment, which can increase risk of mortality. Therefore, policy makers should take into consideration patient accessibility, in particular for patients living in vulnerable areas so that medical resources can be more evenly distributed across regions. In addition, health gaps between regions must be continuously considered through routine evaluation of health outcomes between regions.

Reviewer 2 Report

In their retrospective study the authors used the National Health Insurance (NHI) Cohort Database with the aim of the invetigating the association
between delays in surgical treatment and five- and one- year mortality in patients with lung or gastric cancer. This is a somewhat interesting study focusing on a significant aspect of the treatment of patients with 2 mainstream types of cancer and how it critically affects overall survival. Overall this is a well-structured and written manuscript with minor spelling and languange corrections to be applied. The methods and results are clearly described. The outcomes are adequatelly discussed in the context of the existing literature.

Minor points:

- the abstract needs to be revised; please include full words for NIH as well as include more details on which population it refers to.

Author Response

First, we greatly appreciate the comments and suggestions offered by the reviewers, which we used to improve the manuscript. Our response to each comment follows, and we have attached a revision note and also highlighted the revised sections of the manuscript. Again, thank you for the valuable and helpful comments.

Answer to Reviewer 2:

Minor points:

- the abstract needs to be revised; please include full words for NIH as well as include more details on which population it refers to.

 Answer: Thank you for your comment. We have revised the abstract per your comments as the following(page 1 line 12-13):

The National Health Insurance claims data, 2006 to 2015 were used.

Reviewer 3 Report

Thank you for the opportunity to review this interesting manuscript.  Overall the details were presented coherently to answer the aim of the proposed study. In addition to correcting the formatting by removing unnecessary hypens (e.g. ‘av-erage’ on line 34, ‘Alt-hough’ on line 37, etc) throughout the manuscript, I have a few comments for the authors to consider:

  • Abstract:
    • Line 23: “The findings were not significant for gastric cancer based on 30 days.” Please include the HR and 95% CI. Please also clarify if it was ‘within 30days’ or ‘at 30 days’.
    • Line 24-25: “The trends shown in lung cancer patients were greater in non-capital area.” How about results of other subgroups analyses? And the results for gastric cancer patients? If reporting these exceeded the word limit, the authors may choose to delete the sentence from the abstract since it is not the main research question.
  • Introduction:
    • Please add the rationale of focusing on subgroup analyses based on patient morbidity, type of treatment received, location of institution. Was there a gap of knowledge around the understanding of these factors? Please note that without the rationale, subgroup analyses are often only performed when there was a significant interaction between the key exposure factor (e.g. treatment) and the characteristics based on results from the statistical analyses.
  • Materials and Methods:
    • Line 66: “This data includes …”. Please note that data indicate plural form so please delete the ‘s’ from ‘includes’. Please also correct this throughout the manuscript.
    • Line 70: “…, patients diagnosed before 2004 or those 70 diagnosed with other cancers in the last five years were excluded”. The rationale stated by the authors (line 69) is not relevant in excluding patients diagnosed before 2004, unless there were concerns around the quality of the retrospective data (medical records). If so, please clarify this for the reader, and acknowledge the potential bias in the discussion section.
    • Line 73-74: “Individuals who died within 30 days of first diagnosis were excluded to avoid bias regarding the time to event outcome”. Please clarify the specific bias the authors wanted to avoid. The rationale to exclude them might be due to the diagnosis of a later stage of cancer? If so, the authors could have stated so, and acknowledge the limited generalisability (i.e. not applicable to patients diagnosed with the final phase of cancer) in the discussion section. Excluding these patients might also be another issue (not right censoring but left truncation), which needs to be acknowledged.
    • Line 88-89: Please provide a reference to justify the reason of using those specific (and different) cut-off points of 50 and 30 days.
    • Line 114-115: Please provide a rationale to perform subgroup analyses (see comment from the Abstract above), as well as comparison of region and study period (line 116).
    • Line 116: Please indicate if the significance level was set at 0.05 or 0.01 whilst considering the multiplicity issue.
  • Results:
    • Table 1: The p-values could be revised to include only a maximum of 3 decimal places. Please include a footnote stating the statistical analyses performed to derive the p-values.
    • Figure 1: Please correct the legend as both graphs indicated that they were ‘gastric cancer’, and cross-check/correct the cut-off points – it should be within (and after) 30 days for one and within (and after) 50 days for another cancer. Also, if the authors intend to report the days, the x-axes should be based on days, not months. Alternatively, the authors could leave the x-axes in months but update the survival time on the graph to months instead of days.
    • Table 2: Do clarify if it was the 5-year survival in the title of Table 2. Also clarify if the HR were the adjusted HRs from the Cox models – include a footnote stating the statistical analyses performed to derive the statistics, and outline the adjusted variables. Please also revise the p-values to include only a maximum of 3 decimal places.
    • Figure 2 can be deleted (and the related descriptions of Figure 2 in the Results section, and any related discussion) since the authors had not provided a rationale about the need of performing subgroup analyses.
    • Figure 3: Please clarify if the numbers included in the top tables were mean days or median days, and indicate in the legends that those categories were days.
    • Line 150: “…the degree of differences between regions was also more prominent.” It is not clear if it was more prominent when comparing the study period or the type of cancer. Please clarify. The authors could also state that time to treatment appears to have reduced in 2010-2015 compared to 2005-2009 for both cancers, instead of attempting to hint on any statistical differences.
  •  Discussion:
    • Please include the limitations noted above in the last paragraph of the discussion section. The authors could also include some suggestions to address these issues for future studies.

Author Response

First, we greatly appreciate the comments and suggestions offered by the reviewers, which we used to improve the manuscript. Our response to each comment follows, and we have attached a revision note and also highlighted the revised sections of the manuscript. Again, thank you for the valuable and helpful comments.

Answer to Reviewer 3:

Abstract:

Line 23: “The findings were not significant for gastric cancer based on 30 days.” Please include the HR and 95% CI. Please also clarify if it was ‘within 30days’ or ‘at 30 days’.

Line 24-25: “The trends shown in lung cancer patients were greater in non-capital area.” How about results of other subgroups analyses? And the results for gastric cancer patients? If reporting these exceeded the word limit, the authors may choose to delete the sentence from the abstract since it is not the main research question.

 Answer: Thank you for your comment. We have revised the abstract based on your comments (page 1 line 11-24)

The aim of this study is to investigate the association between delays in surgical treatment and five- and one- year mortality in patients with lung or gastric cancer. The National Health Insurance claims data, 2006 to 2015 were used. The association between time to surgical treatment, in which the cut-off value was set at average time time(50 or 30 days), and five - year mortality was analyzed using the Cox proportional hazard model. Subgroup analysis was performed based on treatment type and location of medical institution. A total of 810 lung and 2,659 gastric cancer patients were included, in which 74.8% of lung and 71.2% of gastric cancer patients received surgery within average. Compared to lung cancer patients who received treatment within 50 days, the five-year (HR 1.826, 95% CI 1.437 - 2.321) mortality of those who received treatment afterwards was higher. The findings were not significant for gastric cancer based on the after 30 days standard (HR: 1.003, 95% CI: 0.822 – 1.225). In lung cancer patients, time-to-treatment and mortality risk were significantly different depending on region. Delays in surgical treatment were associated with mortality in lung cancer patients. The findings imply the importance of monitoring and assuring timely treatment in lung cancer patients.

Introduction:

Please add the rationale of focusing on subgroup analyses based on patient morbidity, type of treatment received, location of institution. Was there a gap of knowledge around the understanding of these factors? Please note that without the rationale, subgroup analyses are often only performed when there was a significant interaction between the key exposure factor (e.g. treatment) and the characteristics based on results from the statistical analyses.

 Answer: Thank you for your comment. We changed our subgroup analysis based on comments from reviewer #1. The subgroup analysis was conducted according to type of treatment and the location of the hospital visited for the initial treatment. In addition, we added the following in the introduction to give an explanation about the need for performing a subgroup analysis as following (page 2 line 54-68):

The healthcare delivery system of Korea encompasses primary, secondary, and tertiary hospitals, with relatively large hospitals being largely concentrated in the capital area. Differences in access to care may be one of the many factors that affect patient care, and patients living away from hospitals report having worse outcomes, including mortality [12, 13]. This means that region, such as urban or rural areas, can affect the timing of treatment. In addition, patient outcome may differ depending on the type of treatment they receive. Therefore, a need exists to evaluate the impact of treatment delay on patient outcomes based on the location of the medical institution and the type of treatment received.

In this study, the association between time to surgical treatment and mortality in patients with gastric or lung cancer was investigated. We hypothesized that delayed treatment can reduce survival, especially in lung cancer patients with poor prognosis. Furthermore, a subgroup analysis was performed to evaluate the impact of treatment type and the location of the medical institution first visited for treatment on survival rate as these factors can affect patient outcomes.

Materials and Methods:

Line 66: “This data includes …”. Please note that data indicate plural form so please delete the ‘s’ from ‘includes’. Please also correct this throughout the manuscript.

 Answer: Thank you for your comments. We have revised this throughout the manuscript.

Line 70: “…, patients diagnosed before 2004 or those 70 diagnosed with other cancers in the last five years were excluded”. The rationale stated by the authors (line 69) is not relevant in excluding patients diagnosed before 2004, unless there were concerns around the quality of the retrospective data (medical records). If so, please clarify this for the reader, and acknowledge the potential bias in the discussion section.

 Answer: Thank you for your comments.

What we excluded from the three-year period is associated with the characteristics of the data used. Our study aimed to evaluate the patient outcome (death) over time from the date of initial diagnosis of cancer to surgery. Hence, a clear definition of the date of initial diagnosis is required. For example, consider the following situation. Patient A had already been diagnosed and treated for cancer in 2001, and cancer recurred in 2002. Also, like patient B, the initial diagnosis was December 2001 and it is possible that the second visit was made in January 2002. Since our data is based on the patient's medical history, to clarify the criteria for initial diagnosis, we washed out three years. Since 2005 the included patients have been defined as a newly diagnosed case.

We have added a flow-diagram to the Appendix to clarify the population selection process. Also, we have revised the Methods section as follows to help readers better understand (page2 line 76-86).

A total of 27,579 patients diagnosed with gastric or lung cancer were included at baseline. A wash out period of three years was applied to clearly define the date of initial cancer diagnosis. In addition, patients who were diagnosed with other cancer types during the follow-up period were excluded. In this process, 8,340 patients were excluded. To reduce heterogeneity among patients, only cancer patients who received surgical treatment within one year of diagnosis were included. Individuals who died within 30 days of first diagnosis were excluded to avoid bias regarding the time to event outcome. Elderly patients over 80 years of age or Medical-Aid beneficiaries were also excluded. This led to the final study population of 810 and 2,659 patients who received surgical treatment for lung or gastric cancer (Appendix figure 1).

Line 73-74: “Individuals who died within 30 days of first diagnosis were excluded to avoid bias regarding the time to event outcome”. Please clarify the specific bias the authors wanted to avoid. The rationale to exclude them might be due to the diagnosis of a later stage of cancer? If so, the authors could have stated so, and acknowledge the limited generalisability (i.e. not applicable to patients diagnosed with the final phase of cancer) in the discussion section. Excluding these patients might also be another issue (not right censoring but left truncation), which needs to be acknowledged.

 Answer: Thank you for your comment.

In our study, patient mortality was evaluated as the outcome variable. We evaluated the effect of patients' average waiting time to surgery (50 days and 30 days) on mortality. If a patient dies within 30 days of diagnosis of cancer, it is more likely that the patient died due to comorbidities or the effect of cancer, rather than the patient's waiting time being a problem that affect patient outcome. We excluded patients within 30 days of diagnosis to exclude the possible effects that may affect mortality other than waiting time.

Line 88-89: Please provide a reference to justify the reason of using those specific (and different) cut-off points of 50 and 30 days.

 Answer: Thank you for your comment. The time we used (30 or 50 days) was set as the cutoff value because it was the average treatment time measured in the study. Depending on the cutoff value, the effect of time-to-treatment on death may appear differently, and this was added to the discussion as a limitation (page 9 line 164-165).

Finally, the study results may be affected by the cutoff criterion and further studies that consider various waiting times are needed.

Line 114-115: Please provide a rationale to perform subgroup analyses (see comment from the Abstract above), as well as comparison of region and study period (line 116).

 Answer: Thank you for your comment. We added the reason for the subgroup analysis in the introduction (page 2 line 54-68):

The healthcare delivery system of Korea encompasses primary, secondary, and tertiary hospitals, with relatively large hospitals being largely concentrated in the capital area. Differences in access to care may be one of the many factors that affect patient care, and patients living away from hospitals report having worse outcomes, including mortality [12, 13]. This means that region, such as urban or rural areas, can affect the timing of treatment. In addition, patient outcome may differ depending on the type of treatment they receive. Therefore, a need exists to evaluate the impact of treatment delay on patient outcomes based on the location of the medical institution and the type of treatment received.

In this study, the association between time to surgical treatment and mortality in patients with gastric or lung cancer was investigated. We hypothesized that delayed treatment can reduce survival, especially in lung cancer patients with poor prognosis. Furthermore, a subgroup analysis was performed to evaluate the impact of treatment type and the location of the medical institution first visited for treatment on survival rate as these factors can affect patient outcomes.

In addition, we presented the average days in Figure 1 to assess time delay from diagnosis by region and study period.

Line 116: Please indicate if the significance level was set at 0.05 or 0.01 whilst considering the multiplicity issue.

 Answer: Thank you for your comment. We have revised the manuscript as following (page 3 line 127):

A p-value <0.05 was considered statistically significant.

Results:

Table 1: The p-values could be revised to include only a maximum of 3 decimal places. Please include a footnote stating the statistical analyses performed to derive the p-values.

 Answer: Thank you for your comments. We have revised the table and added footnote.

Figure 1: Please correct the legend as both graphs indicated that they were ‘gastric cancer’, and cross-check/correct the cut-off points – it should be within (and after) 30 days for one and within (and after) 50 days for another cancer. Also, if the authors intend to report the days, the x-axes should be based on days, not months. Alternatively, the authors could leave the x-axes in months but update the survival time on the graph to months instead of days.

 Answer: Thank you for your comment.

We have revised gastric and lung cancer. Figure 2 shows the Kaplan-Meier survival curve results, including the group of patients who underwent surgery before and after the average number. In Figure 2, red indicates patients who received surgery before the average day (50 or 30 days) and blue indicates patients who received surgery after the average day. We followed all patients for up to 5 years and the 'month' shown on the X-axis is the result reflecting 60 months.

Table 2: Do clarify if it was the 5-year survival in the title of Table 2. Also clarify if the HR were the adjusted HRs from the Cox models – include a footnote stating the statistical analyses performed to derive the statistics, and outline the adjusted variables. Please also revise the p-values to include only a maximum of 3 decimal places.

 Answer: Thank you for your comment. We have revised the table and added footnote.

Figure 2 can be deleted (and the related descriptions of Figure 2 in the Results section, and any related discussion) since the authors had not provided a rationale about the need of performing subgroup analyses

 Answer: Thank you for your comments. We added reasons for performing a subgroup analysis in the introduction section and related its content in the discussion section (page 9 line 225-255):

The results of the subgroup analysis reveal that type of treatment and the of the visiting medical institution can influence patient outcome regarding lung cancer. Lung cancer survival is largely determined by stage and treatment characteristics. Surgical resection is commonly conducted in lower stage patients without metastasis whilst later stage operable patients often receive pre- or post- chemo and or radiotherapy [22]. Previous studies have report that delayed resections may reduce survival rates [7] and hence suggest the importance of timely surgery [23]. Our results also suggest that it is important to receive surgery on time, especially if only surgical treatment is needed, as timely treatment may affect patient outcomes.

In addition, the effect of time to surgical treatment on mortality was more pronounced in patients who received treatment in the non-capital area. This phenomenon can be considered in two aspects. First, the regional imbalance of medical resources must be taken into account [24]. In Korea, most tertiary hospitals are concentrated in the capital area, which are equipped with various professional staff and high-quality radiotherapy facilities and provide high-quality healthcare services [25]. Such tendencies can act affect the patient's choice, which is the second aspect. Since no strong referral system is in place, patients prefer relatively large hospitals based on individual beliefs [26]. In fact, studies show that the number of surgeries performed for lung cancer has increased in large cities where tertiary hospitals are located [25]. In particular, 60% of lung cancer surgeries were conducted in tertiary hospitals located in the capital area and most patients were receiving radiation therapy in Seoul [25, 27]. In terms of healthcare resources, well-equipped medical equipment and physician volume can lead to better patient outcomes [28], which in turn can affect mortality rates.

These results suggest that while differences in treatment time between regions have decreased over the past decade, health disparities still exist as a potential problem. In particular, differences in access to care across regions can lead to delays in treatment, which can increase risk of mortality. Therefore, policy makers should take into consideration patient accessibility, in particular for patients living in vulnerable areas so that medical resources can be more evenly distributed across regions. In addition, health gaps between regions must be continuously considered through routine evaluation of health outcomes between regions.

Figure 3: Please clarify if the numbers included in the top tables were mean days or median days, and indicate in the legends that those categories were days.

 Answer: Thank you for your comment. We have revised figure legends as following (page 5, line 155).

Figure 1. Average time-to-treatment of the study participants by region (days).

Line 150: “…the degree of differences between regions was also more prominent.” It is not clear if it was more prominent when comparing the study period or the type of cancer. Please clarify. The authors could also state that time to treatment appears to have reduced in 2010-2015 compared to 2005-2009 for both cancers, instead of attempting to hint on any statistical differences.

 Answer: Thank you for your comment. We changed Figure 3 to 1, which does not include the analysis taking into account statistical flow, and the manuscript was revised as follows (page 5 line 147-153):

Figure 1 presents the national trend of time-to-treatment in the study participants. The average period of time-to-treatment in lung cancer patients was shortest in patients living in the capital area (45.7 days) and longest in patients at other areas (54.2 days). Over the past 10 years, the average time-to-treatment for lung cancer patients decreasing in all regions. In the case of gastric cancer, unlike lung cancer, the average period of time-to-treatment was longest in the capital area. Overall, treatment time has declined in the past 10 years.

Discussion:

Please include the limitations noted above in the last paragraph of the discussion section. The authors could also include some suggestions to address these issues for future studies.

Answer: Thank you for your comment. We revised our manuscript per your comments.

Round 2

Reviewer 1 Report

The paper by Kim et al. has been reviewed correctly and it has improved as requested.

In my opinion, now the paper can be published without any further revision.